# Virtual-freezing fluorescence imaging flow cytometry

Hideharu Mikami [1✉], Makoto Kawaguchi[1], Chun-Jung Huang[1,2], Hiroki Matsumura[1], Takeaki Sugimura[1,3,4], Kangrui Huang[1], Cheng Lei[1], Shunnosuke Ueno[1], Taichi Miura[1], Takuro Ito[1,3], Kazumichi Nagasawa[5], Takanori Maeno[5], Hiroshi Watarai[5,6], Mai Yamagishi [7], Sotaro Uemura[7], Shinsuke Ohnuki[8], Yoshikazu Ohya[8,9], Hiromi Kurokawa[10], Satoshi Matsusaka[10,11], Chia-Wei Sun[2], Yasuyuki Ozeki [12✉] & Keisuke Goda[1,3,13,14✉]

By virtue of the combined merits of flow cytometry and fluorescence microscopy, imaging flow cytometry (IFC) has become an established tool for cell analysis in diverse biomedical fields such as cancer biology, microbiology, immunology, hematology, and stem cell biology. However, the performance and utility of IFC are severely limited by the fundamental trade-off between throughput, sensitivity, and spatial resolution. Here we present an optomechanical imaging method that overcomes the trade-off by virtually freezing the motion of flowing cells on the image sensor to effectively achieve 1000 times longer exposure time for microscopy-grade fluorescence image acquisition. Consequently, it enables high-throughput IFC of single cells at >10,000 cells s$^{-1}$ without sacrificing sensitivity and spatial resolution. The availability of numerous information-rich fluorescence cell images allows high-dimensional statistical analysis and accurate classification with deep learning, as evidenced by our demonstration of unique applications in hematology and microbiology.

[1] Department of Chemistry, The University of Tokyo, Tokyo 113-0033, Japan. [2] Department of Photonics, National Chiao Tung University, Hsinchu 300, Taiwan. [3] Japan Science and Technology Agency, Saitama 332-0012, Japan. [4] CYBO, Tokyo 101-0022, Japan. [5] Center for Stem Cell Biology and Regenerative Medicine, The University of Tokyo, Tokyo 108-8639, Japan. [6] Department of Immunology and Stem Cell Biology, Faculty of Medicine, Kanazawa University, Ishikawa 920-8640, Japan. [7] Department of Biological Sciences, The University of Tokyo, Tokyo 113-0033, Japan. [8] Department of Integrated Biosciences, Graduate School of Frontier Sciences, The University of Tokyo, Kashiwa 277-8562, Japan. [9] AIST-UTokyo Advanced Operando-Measurement Technology Open Innovation Laboratory (OPERANDO-OIL), National Institute of Advanced Industrial Science and Technology (AIST), Chiba 277-8565, Japan. [10] Department of Clinical Research and Regional Innovation, Faculty of Medicine, University of Tsukuba, Ibaraki 305-8577, Japan. [11] Department of Gastroenterology, Cancer Institute Hospital, Japanese Foundation for Cancer Research, Tokyo 135-8550, Japan. [12] Department of Electrical Engineering and Information Systems, The University of Tokyo, Tokyo 113-8656, Japan. [13] Institute of Technological Sciences, Wuhan University, Hubei 430072, China. [14] Department of Bioengineering, University of California, Los Angeles, CA 90095, USA. ✉email: mikami@chem.s.u-tokyo.ac.jp; ozeki@ee.t.u-tokyo.ac.jp; goda@chem.s.u-tokyo.ac.jp

Over the last decade, imaging flow cytometry (IFC)[1–9] has opened a new window on biological and medical research by offering capabilities that are not possible with traditional flow cytometry. IFC provides quantitative image data of every event (e.g., cells, cell clusters, debris), allowing the morphometric characterization of single cells in large heterogeneous populations[1,2] and further advancing our understanding of cellular heterogeneity[10]. Furthermore, conventional digital analysis tools in flow cytometry such as histograms and scatter plots are readily available to IFC users, but with much richer information about the acquired events by virtue of the image data[1,2]. The availability of the big data produced by IFC is well aligned with the pressing need for progressively larger biomedical datasets for efficient and accurate data analysis with the help of machine learning (e.g., deep learning) to make better decisions in biomedical research and clinical diagnosis[11,12]. Recent studies show that IFC is highly effective for the localization and enumeration of specific molecules such as proteins, nucleic acids, and peptides[3,6], the analysis of cell-cell interaction[13] and cell cycle[12,14], the characterization of DNA damage and repair[15], and fluorescence in situ hybridization (FISH)[16].

Unfortunately, the performance and utility of IFC are constrained by the fundamental trade-off between throughput, sensitivity, and spatial resolution[17–22]. As the flow speed is increased for higher cell throughput, the integration time of the image sensor is inevitably required to be shorter for blur-free image acquisition. A detrimental consequence of this effect is reduction in sensitivity or need for decreased pixel resolution to compensate for the reduced sensitivity. To circumvent this problem, time delay and integration (TDI) with a charged coupled device (CCD) image sensor has been introduced into IFC by Amnis Corporation[1,2]. The TDI is based on the accumulation of multiple exposures of a flowing cell with multiple rows of the CCD's photo-sensitive elements by synchronizing its motion by charge transfer with the exposures[1,2,19], but sacrifices throughput due to the limited readout rate of the CCD (up to ~100 MS s$^{-1}$), resulting in a throughput of 100–1000 cells s$^{-1}$ (depending on the required spatial resolution)[1,2,19], which is 10–100 times lower than that of conventional non-imaging flow cytometry. Furthermore, the CCD suffers from large readout noise (typically tens of photoelectrons), limiting its detection sensitivity. Another approach to overcoming the trade-off is single-pixel imaging that achieves both high-throughput and high-spatial resolution[20–22], but comes at the expense of sensitivity. Also, a combination of parallelized microchannels, stroboscopic illumination, and image acquisition with a high-speed complementary metal oxide semiconductor (CMOS) image sensor has been demonstrated to achieve high-throughput IFC[18], but also suffers from low sensitivity. A common trait of these techniques is the compromise of one of the key parameters in favor of the others, hence limiting the utility of IFC to niche applications.

Our optomechanical imaging method, which we refer to as virtual-freezing fluorescence imaging (VIFFI), overcomes the above trade-off and hence achieves high-throughput (>10,000 cells s$^{-1}$), high-spatial resolution (~700 nm), and high sensitivity simultaneously when combined with IFC (namely, VIFFI flow cytometry). Specifically, the method is based on the principle that the motion of a flowing cell is virtually "frozen" on an image sensor by precisely canceling the motion with a flow-controlled microfluidic chip, a speed-controlled polygon scanner, and a series of timing control circuits in order to increase the exposure time of the image sensor and form a fluorescence image of the cell with significantly high signal-to-noise ratio (SNR). Two additional yet essential elements of VIFFI flow cytometry that maximize the effect of virtual freezing are a light-sheet excitation beam scanner that scans over the entire field of view (FOV)

during the exposure time of the image sensor and the precise synchronization of the timings of the image sensor's exposure and the excitation beam's illumination and localization with respect to the rotation angle of the polygon scanner. As a result of combining these elements, our virtual-freezing strategy effectively enables 1000 times longer signal integration time on the image sensor, far surpassing previous techniques[23,24], and hence achieves microscopy-grade fluorescence imaging of cells at a high flow speed of 1 m s$^{-1}$.

## Results

**Schematic of VIFFI flow cytometry.** As schematically shown in Fig. 1a (see Supplementary Fig. 1 for a detailed schematic), our VIFFI flow cytometer consists of (i) a flow-controlled microfluidic chip, (ii) a light-sheet optical excitation system composed of excitation lasers (Nichia NDS4216, $\lambda = 488$ nm, MPB Communications 2RU-VFL-P-5000-560-B1R, $\lambda = 560$ nm, see Methods), an excitation beam scanner (acousto-optic beam deflector, ISOMET OAD948), and a cylindrical lens ($f = 18$ mm), (iii) a scientific complementary metal–oxide–semiconductor (sCMOS) camera (PCO edge 5.5), and (iv) an optical imaging system composed of an objective lens (NA = 0.75, 20×), a 28-facet polygon scanner as a fluorescence beam scanner (Lincoln Laser RTA-B) whose mirror facets are placed in the Fourier plane of fluorescence from the flowing cells, two relay lens systems with magnifications of 0.2 and 5, each of which is composed of four achromatic lenses, designed to meet the facet size and avoid unwanted aberrations, and a tube lens ($f = 180$ mm) to form a wide-field fluorescence image of the flowing cells on the camera (see Methods, Supplementary Figs. 2–6 for more details of the design). The polygon scanner is used to cancel the flow motion of the flowing cells on the camera by rotating the scanner in the opposite direction to the flow at the angular speed corresponding to the flow and hence to produce a blur-free fluorescence image of the virtually stationary cells on the camera (Fig. 1a and Supplementary Movie 1). However, since this motion cancellation strategy itself is insufficient for microscopy-grade imaging of cells in a high-speed flow due to the residual motion blur caused by the longitudinal variations of the flow speed of cells and aberration (image distortion) in the imaging system that inevitably arise in a significantly large FOV, the localized (but highly efficient) light-sheet excitation beam is scanned in the opposite direction to the flow to illuminate the entire FOV while limiting the local exposure time to 10 μs [given by the beam size (26 μm) divided by the relative scan speed of the excitation beam (2.54 m s$^{-1}$), see Methods], which greatly relaxes the requirement for the flow speed precision and reduces the aberration in the imaging system for blur-free image acquisition in the VIFFI flow cytometer (Fig. 1b). As a result, much longer integration time on the camera can be achieved as long as the fluorescence from the cells stays in a single mirror facet of the polygon scanner (see Supplementary Fig. 2). The scanner keeps rotating during the image data transfer of the camera so that its next facet comes to the Fourier plane of the fluorescence from the cells when the next frame of the camera starts (see Methods for more details). This sequence is carefully designed and optimized such that the camera's neighboring frames have a slightly overlapped imaging area in order to exclude a dead imaging area, accommodate randomly distributed cells in line (which are subject to Poisson statistics), and maximize the throughput of the VIFFI flow cytometer (Fig. 1c). All the frames are continuously acquired at a frame rate of 1,250 fps and stored on a solid-state drive up to 1 TB (0.44 MB frame$^{-1}$ × 2,270,000 frames), allowing a non-stop image acquisition of up to 18,000,000 cells at 10,000 cells s$^{-1}$ (Supplementary Fig. 5).

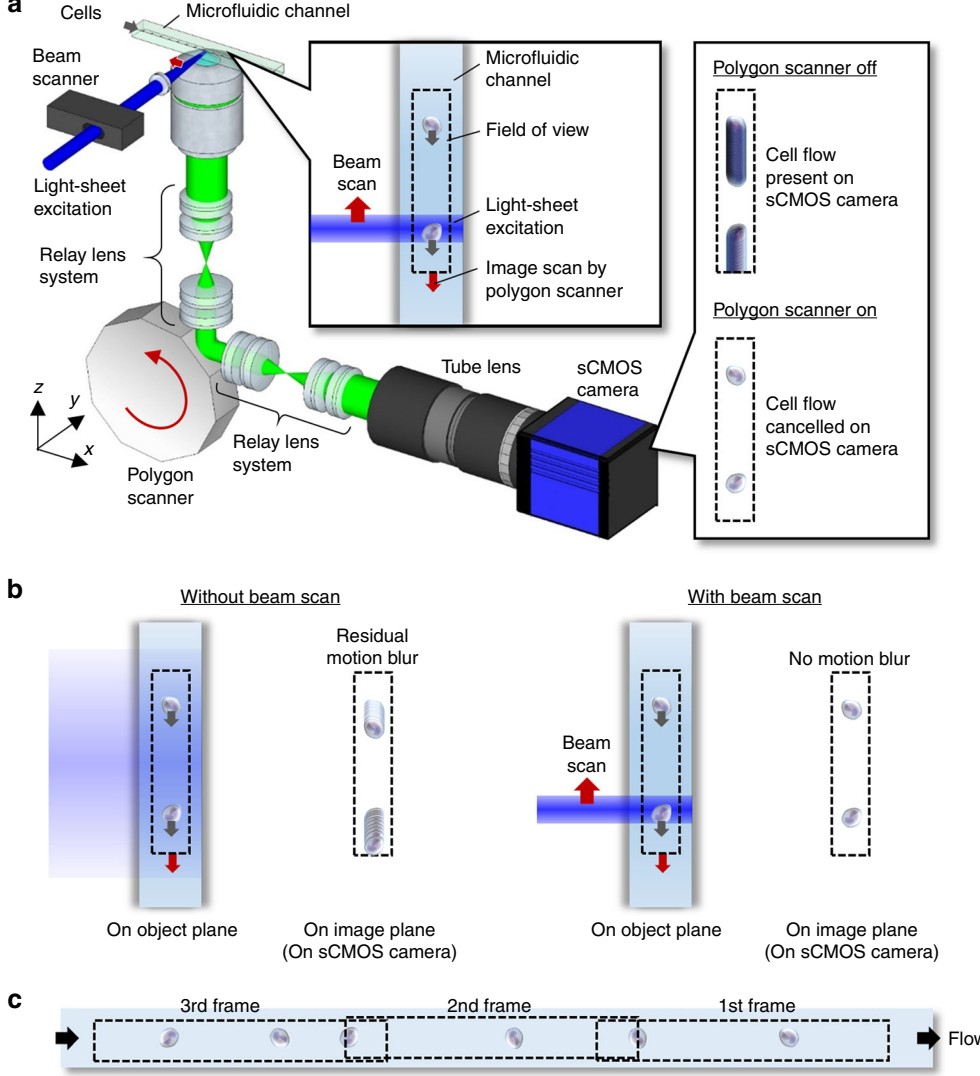

**Fig. 1 VIFFI flow cytometry. a** Schematic and functionality. The side inset shows the effect of VIFFI on the formed images of cells on the sCMOS camera. The upper inset shows scanned light-sheet excitation of cells flowing in the microfluidic channel. The motion of a flowing cell is virtually "frozen" on the camera by precisely canceling the motion in order to increase the exposure time of the image sensor and form a fluorescence image of the cell with 1000-fold increase in the camera's integration time. **b** Comparison in image acquisition on the sCMOS camera with and without the beam scan. **c** Serial image frames acquired by the sCMOS camera.

**Characterization of VIFFI flow cytometry**. As a proof-of-principle demonstration, we used the VIFFI flow cytometer to perform sensitive blur-free fluorescence imaging of fast-flowing biological cells that would be too dim to visualize without VIFFI. Without VIFFI, the camera's exposure time is only limited to 0.3 μs given by the pixel size (325 nm) divided by the flow speed (typically >1 m s$^{-1}$ for high-throughput operation) in order to obtain a blur-free fluorescence image of fast-flowing cells. Specifically, we used immortalized human T lymphocyte cells (Jurkat) and microalgal cells (*Euglena gracilis*) for the demonstration (Fig. 2a). Figure 2b–d show fluorescence images of the cells obtained by IFC without VIFFI with an exposure time of 0.3 μs, IFC without VIFFI with an exposure time of 340 μs, and IFC with VIFFI with an exposure time of 340 μs, respectively. Here, with an average interval of 50–100 μm between consecutive cells, the flow speed of 1 m s$^{-1}$ corresponds to a throughput of >10,000 cells s$^{-1}$ (as experimentally shown in Supplementary Fig. 7), which is equivalent to the throughput of commercially available non-imaging flow cytometers[25] (while the throughput value in flow cytometry generally depends on various factors such as cell size

and cell concentration). The rotation of cells in the flow is negligible during the exposure time of the camera since the local exposure time of 10 μs is sufficiently short for the potential rotational motion of the cells to occur. It is evident from the comparison of the fluorescence images that VIFFI significantly improved the spatial resolution and SNR in the images without sacrificing the throughput (see Methods, Supplementary Figs. 8 and 9 for more details).

The high sensitivity of VIFFI flow cytometry allows for fluorescence imaging of various types of cells (e.g., cancer cells, microalgal cells, budding yeast cells, white blood cells) flowing at a high speed of 1 m s$^{-1}$ (Fig. 3a through Fig. 3f). For example, the ability to enumerate localized fluorescent spots by FISH imaging (Fig. 3a) indicates its potential application to real-time characterization of gene copy number alterations in circulating tumor cells (CTCs) in blood[17]. Also, it enables precise analysis of the cell cycle of budding yeast (*Saccharomyces cerevisiae*) cells (Fig. 3b). In addition, it is useful for identifying fine morphological and structural features of single cells in large populations, such as the indented elliptical shape of *Chlamydomonas reinhardtii* cells

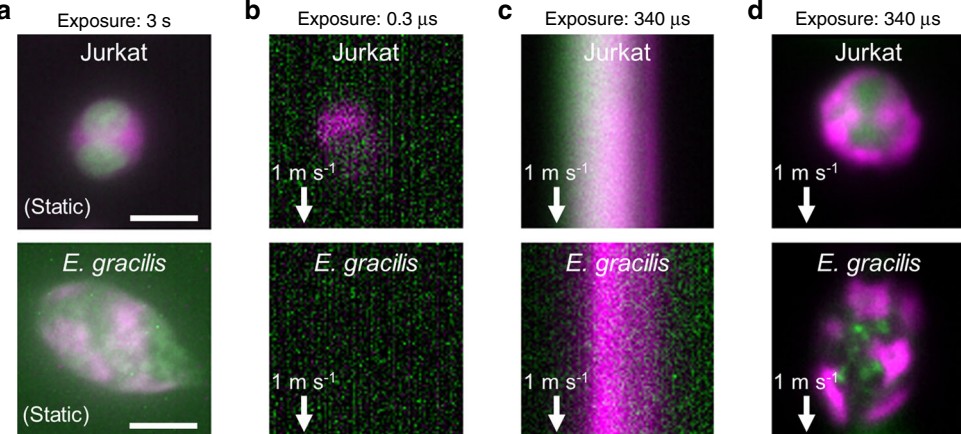

**Fig. 2 Demonstration of VIFFI flow cytometry.** Immortalized human T lymphocyte cells (Jurkat) and *E. gracilis* cells are used for all cases. **a** Fluorescence images of the cells at rest obtained by conventional fluorescence microscopy. The images are representatives of >10 images of cells obtained under identical imaging conditions. **b** Fluorescence images of the cells in a 1-m s$^{-1}$ flow obtained by IFC without VIFFI with an exposure time of 0.3 μs. **c** Fluorescence images of the cells in a 1-m s$^{-1}$ flow obtained by IFC without VIFFI with an exposure time of 340 μs. **d** Fluorescence images of the cells in a 1-m s$^{-1}$ flow obtained by IFC with VIFFI with an exposure time of 340 μs. Green: nucleus for Jurkat cells (stained by SYTO16), lipids for *E. gracilis* cells (stained by BODIPY505/515). Magenta: cytoplasm for Jurkat cells (stained by CellTracker Red), chlorophyll for *E. gracilis* cells (autofluorescence). It is clear from the comparison of the fluorescence images that VIFFI significantly improved the spatial resolution and SNR in the images without sacrificing the throughput. Scale bars: 10 μm.

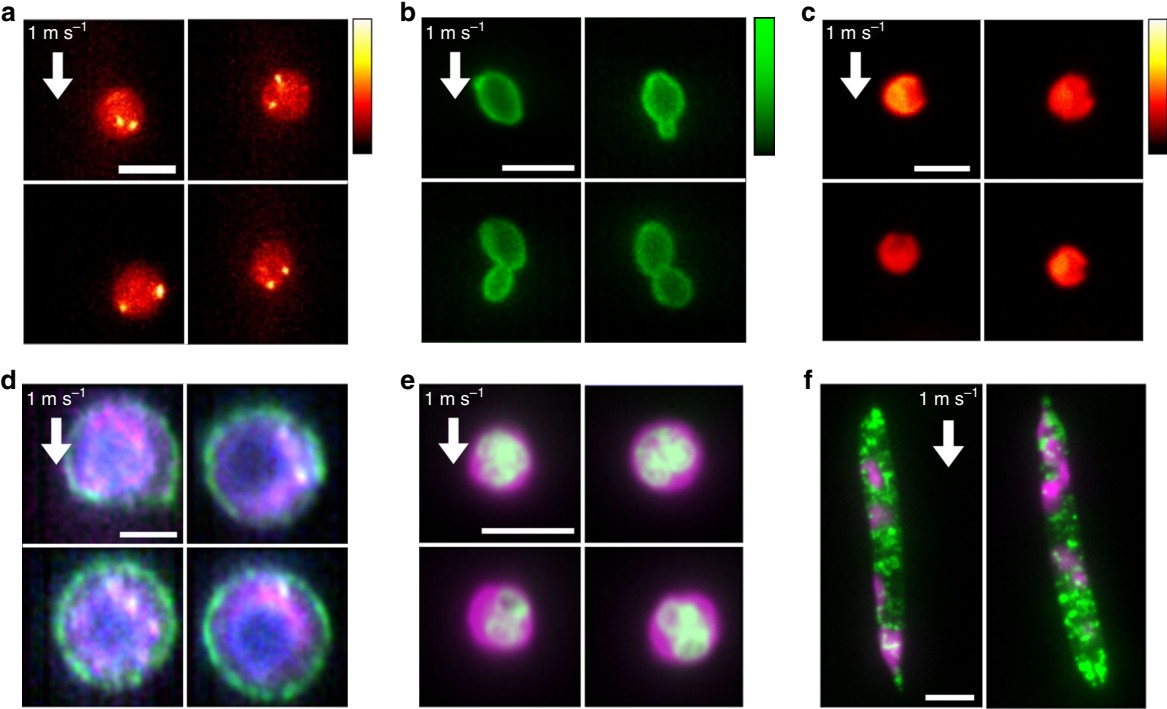

**Fig. 3 Fluorescence images of diverse cell types obtained by VIFFI flow cytometry.** All the images were obtained at a flow speed of 1 m s$^{-1}$. **a** FISH images of Jurkat cells. Two bright spots (shown in yellow-white) corresponding to two copies of chromosome 8 are evident in each cell. **b** Fluorescence images of *S. cerevisiae* whose cell wall was stained by FITC-concanavalin A, showing budding daughter cells from their mother cells. **c** Autofluorescence images of *C. reinhardtii* cells, showing their characteristic morphological features (indented elliptical shape at the head). **d** Three-color fluorescence images of human lung adenocarcinoma cells (PC-9). Magenta: protoporphyrin IX induced by 5-aminolevulinic acid; Green: EpCAM stained by VU-1D9; Blue: nucleus stained by Hoechst 33342. **e** Two-color fluorescence images of murine neutrophils. Green: nucleus stained by SYTO16; Magenta: cytoplasm stained by CellTracker Red. **f** Two-color fluorescence images of *E. gracilis* cells. Green: lipids stained by BODIPY505/515; Magenta: autofluorescence of chlorophyll. Scale bars: 10 μm.

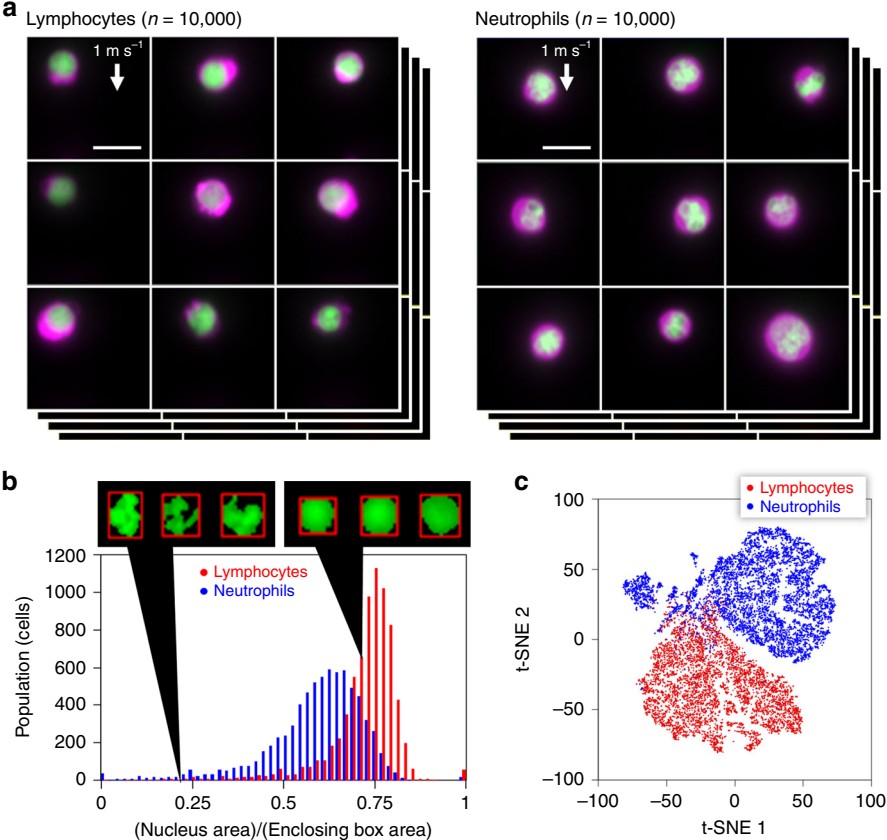

**Fig. 4 High-throughput, high-content screening of murine neutrophils, and lymphocytes with VIFFI flow cytometry. a** Libraries of fluorescence images of the cells ($n = 10,000$ for each group) obtained by VIFFI flow cytometry. The cytoplasm was stained by CellTracker Red (shown in magenta) while the nucleus was stained by SYTO16 (shown in green). Different color scales are used for the fluorescence images of murine neutrophils and lymphocytes for higher image contrast. See Supplementary Figs. 19 and 20 for more images. **b** Histogram of the cells in the ratio in area between the nucleus and enclosing box ($n = 7399$ for lymphocytes and $n = 7398$ for neutrophils). The source data is available in our Source Data file. **c** t-SNE plot of the cells obtained with 4096 features by VGG-16 ($n = 8000$ for each group). A sub-type of neutrophils is evident at around (t-SNE 1, t-SNE 2) = ($-70$, 40). The source data is available in our Source Data file. Scale bars: 10 μm.

(Fig. 3c), the boundary (cell surface) localization of the epithelial cell adhesion molecule (EpCAM) in CTCs (Fig. 3d), nuclear lobulation in murine neutrophils (Fig. 3e), and lipid droplet localization in *E. gracilis* cells (Fig. 3f) that have not been possible with previous high-throughput imaging flow cytometers at this flow speed[20,26] due to their limited imaging sensitivity. Below we used murine white blood cells and *E. gracilis* cells to show practical applications of VIFFI flow cytometry.

**Applications of VIFFI flow cytometry.** One of many applications where VIFFI flow cytometry is effective is to significantly improve statistical accuracy in the identification and classification of white blood cells based on morphological phenotypes (e.g., size, shape, structure, nucleus-to-cytoplasm ratio)—a routine practice for clinical diagnoses in which the cell throughput and hence classification accuracy are limited due to the manual examination of cells under conventional microscopes by skilled operators. Specifically, we used the VIFFI flow cytometer to obtain a large number of high-resolution, high-SNR fluorescence images of murine lymphocytes and neutrophils (Fig. 4a and Supplementary Fig. 10). The images enable the accurate quantification of nuclear lobulation by analyzing the ratio in area between the nucleus and enclosing box (the rectangular box with the smallest area within which the nucleus lies), which effectively brings out the differences between the two types of cells including their distinct heterogeneity in population distribution (Fig. 4b,

Methods). Also, the obtained images quantitatively elucidate morphological features of each cell type such as cell area with high precision (Supplementary Fig. 11). Furthermore, the information-rich cell images allow the use of a deep neural network for cell classification with even higher accuracy. Specifically, we employed a convolutional neural network (CNN) with 16 layers (VGG-16)[27] (see Methods for details of training) and achieved a high classification accuracy of 95.3% between the two populations. A scatter plot of the cells obtained from 4,096 features generated by VGG-16 through t-distributed stochastic neighbor embedding (t-SNE)[28] shows an obvious separation between the populations (Fig. 4c), indicative of the high classification accuracy. The plot also shows the existence of one or more sub-populations among neutrophils, suggesting a possibility that neutrophils can further be classified into sub-types, which potentially has a significant biological implication in the field of immunology[29].

Another unique application of VIFFI flow cytometry is the determination of the population, morphological properties, and spatial distribution of specific intracellular molecules in large heterogeneous cell populations. Specifically, we used the VIFFI flow cytometer to study lipid droplets (i.e., neutral lipid storage sites) within single cells of *E. gracilis*, a unicellular photosynthetic microalgal species known to produce wax esters suitable for biodiesel and aviation biofuel[30]. While recent studies show that lipid droplets play a key role in lipid biosynthesis, mobilization,

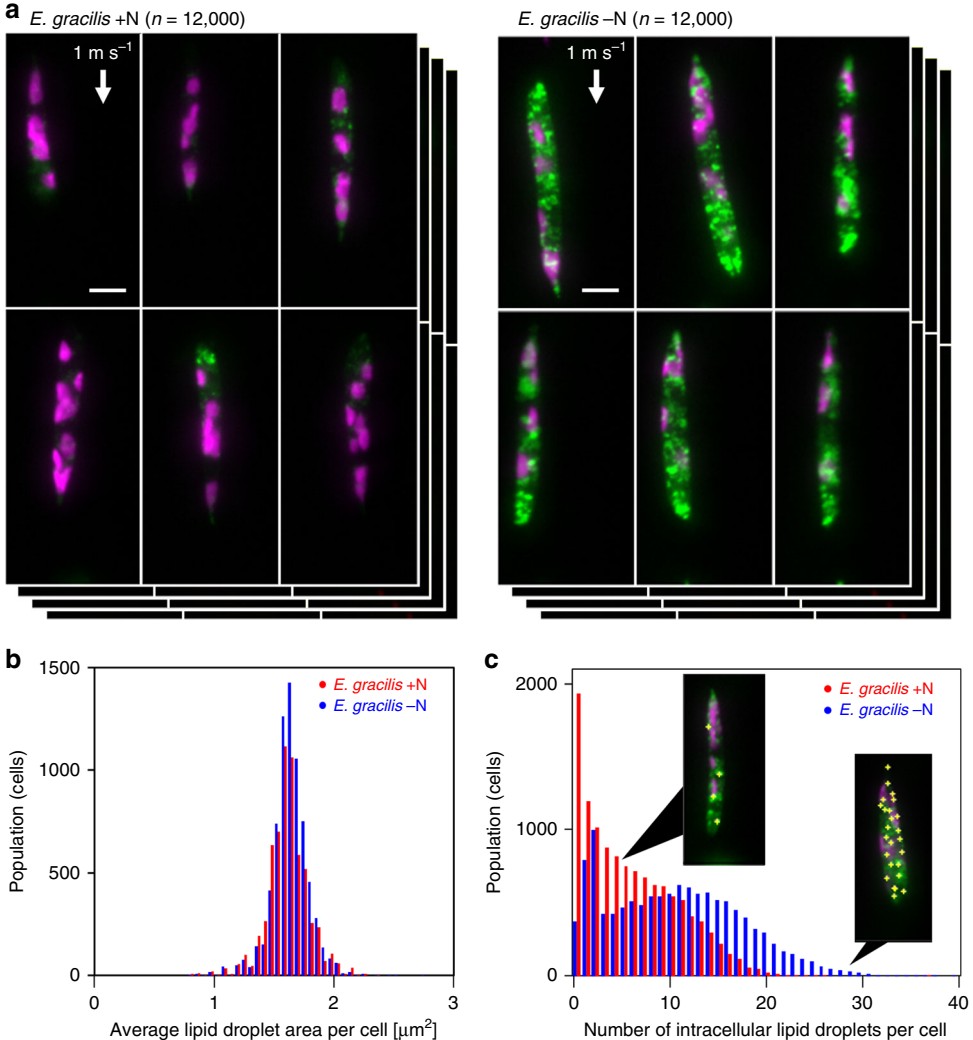

**Fig. 5 High-throughput, high-content screening of *E. gracilis* cells. a** Fluorescence images of nitrogen-sufficient (+N) and nitrogen-deficient (−N) *E. gracilis* cells ($n = 12,000$ for each group) obtained by VIFFI flow cytometry. The lipids (shown in green) were stained by BODIPY505/515 while chlorophyll (shown in magenta) was autofluorescent. See Supplementary Figs. 16 and 17 for more images. **b** Histograms of *E. gracilis* cells in the average droplet area per cell ($n = 7499$ for each group). The source data is available in our Source Data file. **c** Histograms of *E. gracilis* cells in the number of intracellular lipid droplets per cell ($n = 11,999$ for each group). Yellow plus signs in the insets indicate the locations of lipid droplets within the cell body. It is important to note that while the average droplet area does not change under two culture conditions, the number of lipid droplets varies significantly. The near-origin peak of the *E. gracilis* −N histogram is due to isolated lipid droplets, presumably caused by the fracture of cells in the sample preparation process (see Supplementary Fig. 21). The source data is available in our Source Data file. Scale bars: 10 μm.

and homeostasis, little is known about microalgal lipid droplets[31] despite their importance for efficient microalgae-based biofuel production. As shown in Fig. 5a, the VIFFI flow cytometer enables high-throughput single-cell imaging of numerous *E. gracilis* cells under two culture conditions (nitrogen-sufficient and nitrogen-deficient). Here, nitrogen deficiency is an environmental stress condition known to induce microalgae (including *E. gracilis*) to accumulate lipids in the cell body. From the acquired high-resolution, high-SNR images, we were able to localize and enumerate intracellular lipid droplets and quantify their areas (Fig. 5b, c, see Supplementary Fig. 12 for CNN-based classification results) with sub-1 μm resolution and 100 nm precision (see Methods, Supplementary Figs. 13–15). These results indicate that, while there is no significant difference in the average lipid droplet area per cell between the two cultures, there is an obvious difference in the distribution of the number of lipid droplets per cell between them. In addition, we found that the degree of heterogeneity in the cell populations is comparable to or even larger than the overall separation between them. Furthermore, the obtained images (Supplementary Figs. 16 and 17) indicate that the intracellular lipid droplet accumulation occurred non-uniformly, which was more prominent in the cells under the nitrogen-sufficient condition. These results provide an important insight into the lipid biosynthesis of *E. gracilis* and hence efficient metabolic engineering[32].

## Discussion

From a practical perspective, the specifications of the VIFFI flow cytometer demonstrated above can further be boosted by replacing its key components with more advanced commercial products. First, the imaging speed and FOV in the $y$ direction can be improved by replacing the sCMOS camera with a high-speed camera at the expense of imaging sensitivity. Specifically, a throughput of >100,000 cells s$^{-1}$ (provided by a flow speed of 10 m s$^{-1}$) with a FOV of 65 μm in the $y$ direction can be achieved

by employing a commercial high-speed camera (Photron FAS-TCAM Mini WX100). Second, the imaging sensitivity can be increased by replacing the sCMOS camera with a newly released camera with a higher quantum efficiency. In fact, sCMOS cameras with higher sensitivity (e.g., Hamamatsu ORCA-Flash4.0 V3, quantum efficiency ≥ 80% at maximum) than the present one (quantum efficiency = ~60% at maximum) are commercially available. Finally, the overall size of the VIFFI flow cytometer (~1 m$^2$) can be reduced by replacing the relay lens systems with custom lenses, allowing for benchtop implementation.

With the flexibility and scalability of VIFFI flow cytometry, its capabilities can further be enhanced in various directions, thereby extending the range of applications and discoveries which are accessible by them. First, while we used two colors in our proof-of-principle demonstration, fluorescence imaging in several colors can easily be conducted with a proper set of dichroic beams-plitters. Second, VIFFI flow cytometry is essentially compatible with advanced image-sensor-based fluorescence microscopy techniques such as structured illumination microscopy[33], making it feasible to perform super-resolution fluorescence IFC. Third, the flow speed (i.e., throughput) can be increased as long as it obeys the trade-off relation between the flow speed, the FOV in the $y$ direction, and the exposure time, as shown in Supplementary Fig. 3. As the magnification of the fluorescence imaging system also affects the flow speed, a lower magnification value enables higher flow speed at the cost of pixel resolution. Fourth, the recent advancement of machine learning methods will help further enhance the capability of data analysis obtained by VIFFI flow cytometry. Fifth, since the data transfer rate of the sCMOS camera is a performance-limiting factor, future advances of the image sensor technology are expected to further increase the cell throughput and the number of fluorescence colors. Sixth, to avoid image artifacts due to the limited depth of field (e.g., image defocus), an extended depth-of-field (EDF) technique can be implemented in the VIFFI flow cytometer[34], which is particularly useful for FISH imaging and imaging of large cells. Specifically, a cubic phase mask[34] can be inserted in the optical path of fluorescence detection for the EDF. Finally, the full potential of VIFFI flow cytometry can be exploited by incorporating a cell-sorting module and a real-time intelligent image processor into it for intelligent image-activated cell sorting[26,35], which will allow for subsequent detailed analysis of target cells via electron microscopy and DNA/RNA sequencing, as well as for subsequent use of them for synthetic biology via directed evolution.

In addition to the above applications, the high-throughput, high-sensitivity, high-spatial-resolution imaging capability of VIFFI flow cytometry opens a window onto a new class of biological, pharmaceutical, and medical applications. First, as indicated by the images shown in Fig. 3a, it enables high-throughput FISH analysis[16]. FISH has been employed in a wide variety of applications such as diagnosis of acute lymphoblastic leukemia and Down syndrome, identification of bacterial pathogens, and detection of minimal residual disease, but conventional FISH requires microscopy-based observation of cells and hence falls short in screening large populations of cells. Since VIFFI flow cytometry extends the applicability of FISH to such large cell populations (e.g., blood), it is expected to offer a significant impact on pathology and microbiology. Second, as indicated by the images shown in Fig. 3b, VIFFI flow cytometry allows for large-scale analysis of *S. cerevisiae* mutants based on morphological phenotype, thereby offering the ability to uncover its genotype-phenotype relation[36,37], which has been an important subject of research among *S. cerevisiae* researchers worldwide for synthetic biology. Specifically, the high-spatial resolution of VIFFI flow cytometry can accurately quantify and characterize the morphological features (e.g., area, perimeter, aspect ratio) of each

cell and its intracellular organelles (e.g., nuclei, actin). Third, as indicated by the images shown in Fig. 3c, VIFFI flow cytometry can be used for evaluating mutagenesis of microalgal cells. While previously reported high-throughput imaging flow cytometry can identify mutants of *C. reinhardtii* that express de-localized low-CO$_2$ inducible protein B (LCIB) for studying the carbon concentration mechanism in microalgal photosynthesis[26], the high sensitivity of VIFFI flow cytometry allows for more detailed analysis of the LCIB such as granularity, hence assisting microbiologists with investigation of the carbon concentration mechanism[38]. Finally, as indicated by the images shown in Fig. 3d, VIFFI flow cytometry can be used for accurate detection and enumeration of CTCs in large heterogeneous samples of blood cells[39]. Since it can visualize and differentiate single CTCs and CTC clusters and enumerate CTCs in each CTC cluster, it is expected to help cancer biologists study the relation between the cluster size and the metastatic propensity and spread of CTCs[39,40] and unravel tumor heterogeneity in cancer patients by large-scale single-cell profiling of CTCs in blood[41].

## Methods

**Optical design**. The optical system of the VIFFI flow cytometer was designed so that it has both subcellular spatial resolution and high-throughput comparable to conventional (non-imaging) flow cytometers. On the basis of this concept, we firstly chose an objective lens (Olympus UPLSAPO20x, NA = 0.75) and a tube lens (Olympus U-TLU, $f = 180$ mm) to obtain subcellular resolution. Subsequently, we determined the magnification of the relay lens systems $M$ and $M^{-1}$ and the number of facets of the polygon scanner $N$ by considering the following constraints: (i) The image frames should be concatenated so that whole images of all flowing cells fall in one of the continuous frames (Fig. 1c); (ii) The fluorescence beam size in the polygon scanner's facet should be smaller than the facet size (Supplementary Fig. 2a). In addition, we considered specifications of the sCMOS camera and polygon scanner (Lincoln Laser RTA-B) such as the frame period $T_s = 0.8$ ms (frame rate: 1250 fps) with the region of interest of 2560 × 88 pixels, sensor size in the direction corresponding to the cell flow $L_x = 16.6$ mm, and inner diameter of the polygon mirror of the scanner $d_p = 70$ mm. The above constraints are expressed by the following equations:

$$4f_o M \left\{ \frac{2\pi}{N} + \sin^{-1}\left[ \left( -\frac{d_o M}{d_p} + \sin\alpha \right)\cos\frac{\pi}{N} \right] - \sin^{-1}\left[ \left( \frac{d_o M}{d_p} + \sin\alpha \right)\cos\frac{\pi}{N} \right] \right\} > 0, \quad (1)$$

$$\frac{v T_s N}{4\pi f_o} < M < \frac{L_x N}{4\pi f_t}, \quad (2)$$

where $f_o$, $\alpha$, $d_o$, $v$, $f_t$ denote the focal length of the objective lens (9 mm), the nominal incidence angle of the fluorescence to the polygon scanner (45 degrees), the back-aperture diameter of the objective lens (13.5 mm), the flow speed of the cells (1 m s$^{-1}$), and the focal length of the tube lens (180 mm), respectively. The left hand side of Eq. (1) corresponds to the maximum scan range of the polygon scanner on the object and thus represents the maximum exposure duration. A plot of the maximum scan range under the above constraints is shown in Supplementary Fig. 2b. The scan range divided by the flow speed represents the upper limit on the exposure time. The figure shows that a lower magnification value is beneficial for obtaining a longer exposure time. As a trade-off, in practice, a relay lens system with a lower magnification tends to have larger aberration, which degrades the imaging quality. Therefore, we chose $M = 0.2$ and $N = 28$ so that the aberration that occurs in the relay lens systems does not significantly degrade the imaging quality (see below for the detailed design of the relay lens systems) while significant improvement of the imaging sensitivity is obtained.

**Excitation beam scan**. The excitation beam scan is an essential part of the VIFFI flow cytometer to enable its exceptionally long exposure time for fast-flowing cells. A schematic of the beam scan is shown in Supplementary Fig. 4a. A focused excitation beam with a diameter of 26 μm in the cell flow direction is scanned in the direction opposite to the cell flow at 2.54 m s$^{-1}$ (a speed relative to flowing cells), limiting the local exposure time to 10 μs. Since the FOV of the camera in the object plane moves together with the cell flow during the exposure time, the scan range (525 μm) is shorter than the FOV in the flow direction (830 μm), which significantly reduces the off-axis aberration components such as image distortion and field curvature. In particular, the image distortion that occurs between the object and the polygon scanner causes position-dependent errors of the motion cancellation effect of the polygon scanner and hence residual motion of the cell images, which limits extension of the exposure time. We experimentally determined the speed of the residual motion using obtained images of fluorescent particles. The results shown in Supplementary Fig. 4b indicate that the speed range of the residual motion is ±1.5% with the beam scan while it is at least ±6% without

the beam scan. Assuming a typical speed fluctuation of flowing cells in a microfluidic channel of 1.5%, the factors for the extension of exposure time per pixel in the VIFFI flow cytometer are $(0.015 + 0.015)^{-1} \approx 33$ with the beam scan and $(0.06 + 0.015)^{-1} \approx 13$ without the beam scan. Moreover, due to the confined excitation beam width in the flow direction (~3% of the FOV of the sCMOS camera), the excitation efficiency improves by a factor of $1/0.03 \approx 33$ under a certain excitation beam power whereas the total exposure time of the camera is extended by the same factor. As a result, the improvement factor of the imaging sensitivity of the VIFFI flow cytometer is $33 \times 33 \approx 1000$ with the beam scan while it is limited to ~13 without the beam scan.

**Data acquisition sequence**. A schematic of the data acquisition sequence is shown in Supplementary Fig. 5. A trigger signal from a photodetector that indicates the polygon scanner's angle is used to generate external trigger signals for the camera's start of exposure and the waveform generator's signal output. Then, the excitation beam is scanned for ~320 μs during the camera's exposure time (~340 μs), which is set shorter than the upper limit determined by the design of the imaging optical system (420 μs). The camera outputs image data right after its exposure time. We set the total period of the image acquisition procedure to be slightly less than the polygon scanner's scan period (800 μs) by adjusting the number of pixel lines in the $y$ direction in the object plane so that every external trigger signal successfully triggers the camera's exposure start.

**Relay lens systems**. We designed the relay lens systems using OpticStudio (Zemax, LLC). Considering cost effectiveness, our design assumed only off-the-shelf achromatic lenses from major optics companies. Also, we assumed four achromatic doublets for constituting a single relay lens system rather than two to reduce the total aberration. We created a macro that automatically evaluates the total aberration of ~10,000 relay lens systems, each of which has a different combination of the four achromatic doublets with the designated magnification (0.2) and with an optimized configuration found by the optimization function of OpticStudio. Thus, we found that a relay lens system shown in Supplementary Fig. 6a has diffraction-limited imaging performance. We employed this system to both of the relay lens systems in the setup of the VIFFI flow cytometer. We evaluated optical transfer functions (OTFs) of the whole relay system that consists of the two relay lens systems and a polygon scanner using a combined optical model shown in Supplementary Fig. 6b. The OTFs were calculated at different positions in the image plane indicated in the upper part of Supplementary Fig. 6c, considering the dynamic localized illumination of the excitation beam during the rotation of the polygon scanner. As shown in Supplementary Fig. 6c, we confirmed that the relay system does not suffer from significant image degradation over the entire FOV.

**Constructed optical setup**. A complete schematic of the VIFFI flow cytometer is shown in Supplementary Fig. 1. The optical system consists of a light-sheet optical excitation system, an imaging optical system, and an angle detection system for a polygon scanner. In the light-sheet optical excitation system, excitation beams with the desired spatiotemporal profiles in the object plane are created. Two excitation beams from laser sources [Nichia NDS4216 ($\lambda = 488$ nm), MPB Communications 2RU-VFL-P-5000-560-B1R ($\lambda = 560$ nm)] are scanned by a beam scanner (acousto-optic deflector, ISOMET OAD948) and are focused on a microfluidic channel through cylindrical lenses ($f = 80$ mm in the $x$ direction and $f = 18$ mm in the $z$ direction). The designed Gaussian beam diameters ($e^{-2}$ intensity) on the object plane are 26 μm × 5.2 μm and 23 μm × 6.4 μm ($x \times z$, see Supplementary Fig. 1 for the coordinates) for the beams of 488 nm and 560 nm, respectively. Since the beam scanner's deflection angle for a certain frequency of the driving signal is proportional to the wavelength of the incident beam, the scan range for the 488-nm beam, which has the shorter scan range, is set so that it covers the entire FOV. Therefore, the effective exposure time of the 560-nm excitation beam is slightly shorter (488/560 = 86%) than that of the 488-nm excitation beam. In the imaging system, fluorescence images of flowing cells are captured by a sCMOS camera (PCO edge 5.5). Fluorescence signals from the flowing cells are relayed by two relay lens systems with magnifications of 0.2 and 5, respectively, such that images of the cells are formed on the sCMOS camera through a tube lens. The magnification of the imaging system is set to be 20×. A polygon scanner (Lincoln Laser RTA-B) is placed in a conjugate plane of the exit pupil of the objective lens (Olympus UPLSAPO20x, NA = 0.75) that is created between the two relay lens systems. The fluorescence is split into shorter and longer wavelength components by a dichroic mirror (Semrock FF580-FDi01, edge wavelength: 580 nm), each of which forms an image on the sCMOS camera at a different position. The orientation of the sCMOS camera in the optical setup is set by taking its data readout sequence into consideration. Since the sCMOS camera reads out the pixel data line-by-line, the frame rate does not depend on the region of interest in the line direction. Therefore, to operate the camera at the maximum pixel data rate, we set the camera so that the flow direction coincides with the line direction of the camera. In the angle detection system, a timing trigger for synchronized operation of the polygon scanner, beam scanner, and sCMOS camera is generated. A laser beam that reflects at the polygon scanner is focused on a pinhole aperture and is detected by a photodetector. The output voltage signal of the photodetector is used as the timing trigger and then

sent to the external trigger input of a trigger generator (Teledyne LeCroy Wavestation 2052). Trigger signals generated by the trigger generator are sent to the external trigger input of the sCMOS camera and the Waveform generator (RIGOL DG 4202) so that they start exposure and generation of a driving signal for the beam scan.

Depending on the application, different configurations can be employed. For the experiments in Fig. 3c, d and Supplementary Fig. 7, we used a 405-nm excitation laser (Oxxius LBX-405-300-CSB-PP) for obtaining images of *C. reinhardtii* and PC-9 cells. For the experiments in Fig. 3a and Supplementary Fig. 9, we used a 488-nm excitation laser (Coherent Genesis CX 488 STM) for FISH imaging and the evaluation of the imaging sensitivity, respectively, where the laser beam illuminates the microchannel through an objective lens (Leica, 20×, NA = 0.75) with carefully designed Gaussian beam diameters ($e^{-2}$ intensity) in the object plane of 31 μm × 62 μm ($x \times y$, see Supplementary Fig. 1 for the coordinates). We used the same laser for imaging of *S. cerevisiae* cells (Fig. 3b) in the standard illumination configuration shown above.

**Digital image processing and deep learning**. A schematic of our digital image processing is shown in Supplementary Fig. 10. From a raw image frame obtained by image acquisition software (PCO Camware 4.04), we extracted the region of the image of each color channel. Then, we created a binary mask image for each color channel using standard image segmentation methods. Subsequently, morphological features were calculated from masked or non-masked images of each channel. For murine white blood cells, we calculated the cell nucleus area and enclosing box area (the smallest rectangular box area within which the nucleus lies). For *E. gracilis* cells, we obtained the number of lipid droplets by counting local maximum points in the images. In order to count lipid droplets, we first set the approximate lipid droplet size and intensity threshold. After that, we used the approximate lipid droplet size to find the areas with the maximum and minimum intensities. Then, we picked the maximum-intensity areas whose intensity differences with surrounding minimum-intensity areas were larger than the threshold and created a Boolean map, which showed the positions of the lipid droplets. In addition, we used a CNN called VGG-16, which is a well-known CNN model, to differentiate between different cell types in our experiments. We made changes to the input layer of the original VGG-16 model in order to make it applicable to our image data. As shown in Supplementary Fig. 18, the VGG-16 model consists of five convolution segments and a fully connected classifier. Each convolution segment is made of a few convolution layers that extract the features of the image data, and one max pooling layer at the end to reduce the data volume. At the fully connected classifier, 4096 features extracted by the convolution segments were converted to one dimension that provided the classification results. For example, if the input data is a 224 × 224 RGB image, which can be presented as a 224 × 224 × 3 matrix, the first convolutional layer in the first segment (conv1_1) transforms this image into a 224 × 224 × 64 volume. As the image goes through the network, this volume becomes smaller in width and height, but larger in depth. The pooling layer in the fifth segment (pool_5) generates a 7 × 7 × 512 volume. When the image goes to the final segment, the 7 × 7 × 512 volume, which represents all the features of the image with local information is mapped to 4,096 features. After that, the very last layer applies the Softmax function to the data in order to generate a probability distribution. To illustrate the classification accuracy as a scatter plot, the 4096 features were also converted to two meta-features through t-SNE, which is a method to reduce the dimensionality of multi-dimensional data. LabVIEW 2016, Python 3 with Numpy 1.16.4, scikit-image 0.21.2, scikit-learn 0.21.2, matplotlib 3.1.2, PIL 6.2.1, Scipy 1.2.1, and Open-CV 4.1.0 library were used for the image processing. The CNN was constructed with Keras 2.3.1 with TensorFlow as the backend.

**Microfluidic chip**. A commercially available glass microfluidic chip with dimensions of 400 μm × 250 μm (Hamamatsu J12800-000-203) was used for the experiments. The microfluidic chip is capable of hydrodynamic focusing in both the lateral and depth directions. Suspended cells in a glass syringe were introduced into the channel by a syringe pump (Harvard apparatus 11 Elite) at a fixed volumetric flow rate. The sheath fluid was also introduced by the same syringe pump. The ratio of the volumetric flow rates of the sample flow and sheath flow was set to 1:700 (except for the experiments for the evaluation of the spatial resolution of the optical setup; see below for details), corresponding to a sample flow diameter of ~13 μm. Based on the theoretical position-dependent variation of the flow speed of a laminar flow, the error in the flow velocity in the FOV is <1%. We also used a home-made microfluidic chip with dimensions of 200 μm × 200 μm, which was capable of hydrodynamic focusing in both the lateral and depth directions[26], for obtaining FISH images (Fig. 3a) and for the evaluation of the imaging sensitivity (Supplementary Fig. 9), and a polydimethylsiloxane (PDMS)-based microfluidic chip with a hydrodynamic focuser in the $y$ direction for imaging of *C. reinhardtii* cells.

**Preparation of Jurkat cells for two-color imaging**. Jurkat cells were obtained from DS Pharma Biomedical (EC86012803-F0) and cultured in Dulbecco's Modified Eagle Medium (DMEM) with 10% fetal bovine serum (FBS), 1% penicillin streptomycin, 1% non-essential amino acids at 37 °C, and 5% $CO_2$. The cells were placed into Corning® T75 flasks (catalog no. 431464) and allowed to spread for

3 days. They were stained with 10 μM CellTracker Red (Thermo Fisher Scientific) and 1 μM SYTO 16 (Thermo Fisher Scientific) in FBS-free culture media at 37 °C for 45 min. Imaging was performed after washing the cells with phosphate-buffered saline (PBS).

**Preparation of Jurkat cells for FISH imaging[19].** An aliquot of $1 \times 10^7$ Jurkat cells in a round-bottom 2.0-mL microtube was fixed and permeabilized by exposure to two concentrations of Farmer's solution [ethanol/acetic acid = 3:1 (v/v)] in PBS at 4 °C; 30% for 30 min and then 70% for 10 min. Cells were centrifuged at $600 \times g$ for 5 min and washed with 2× saline sodium citrate (SSC). After centrifugation again, the cell pellet was resuspended in the mixture of 14 μL of CEP hybridization buffer, 2 μL of Vysis CEP 8 (D8Z2) SpectrumGreen Probe (Abbott Diagnostics, Lake Forest, IL), and 4 μL of Milli-Q, transferred to a 0.2-mL PCR tube, and then heated for hybridization at 80 °C for 5 min and 42 °C for 2 h. Cell suspension was diluted with 100 μL of 2 × SSC and centrifuged to form a pellet. The pellet was resuspended in 0.4 × SSC containing 0.43% of NP40 detergent and incubated at 72 °C for 2 min. The cells were harvested by centrifugation and then resuspended in 100 μL of 1% PFA in PBS. Before they were loaded into the VIFFI flow cytometer, the cells were stained by 1000× diluted 7-aminoactinomycin (7-AAD) Viability Dye [0.005% (w/v) as the original solution, Beckman Coulter, A07704] for counterstaining.

**Preparation of C. reinhardtii cells.** C. reinhardtii TKAC1017 was obtained from the Tsuruoka, Keio, Algae Collection (TKAC) of T. Nakada in the Institute for Advanced Biosciences at Keio University, Japan. It was cultured in culture flasks (working volume: 20 mL) using a modified AF-6 medium without dissolved carbon source. The culture was maintained at 25 °C and illuminated in a 14:10-h light:dark pattern (~120 μmol photon $m^{-2} s^{-1}$). A group of C. reinhardtii cells was pre-cultured in a modified AF-6 medium. C. reinhardtii cells (~$5 \times 10^6$ cells $mL^{-1}$) in early stationary phase were concentrated to ~$3 \times 10^8$ cells $mL^{-1}$ by centrifugation and immediately used for VIFFI flow cytometry.

**Preparation of S. cerevisiae cells.** S. cerevisiae heterozygous gene deletion mutant of RPC10/rpc10Δ was purchased from EUROSCARF (http://www.euroscarf.de/, accession number: Y22837) and used for VIFFI flow cytometry as a representative of S. cerevisiae cells having a characteristic morphological phenotype of elongated cell shape[36]. Cell culture, fixation, and fluorescent staining were performed according to previously developed methods[37] with some modifications; S. cerevisiae cells were grown at 25 °C in an yeast extract peptone dextrose (YPD) liquid medium containing 1% (w/v) Bacto yeast extract (BD Biosciences, San Jose, CA), 2% (w/v) Bacto peptone (BD Biosciences), and 2% (w/v) glucose. After incubation for 16 h, S. cerevisiae cells in logarithmic phase were fixed in a YPD medium supplemented with formaldehyde (final concentration, 3.7%) and potassium phosphate buffer (100 mM [pH 6.5]) for 30 min at 25 °C. Before VIFFI flow cytometry, cell-surface mannoproteins were stained by 1 mg/mL fluorescein iso-thiocyanate (FITC)-conjugated concanavalin A (Sigma-Aldrich, St. Louis, MO) in P buffer (10 mM sodium phosphate and 150 mM NaCl [pH 7.2]) for 10 min. After washing with P buffer twice, the S. cerevisiae cells were suspended in PBS.

**Preparation of PC-9 cells.** PC-9 (human lung cancer) cells were cultured in a RPMI-1640 medium (Sigma-Aldrich, R8758-500 mL) supplemented with 10% FBS (BOVOGEN, catalog no. SFBS-F lot no. 11555), 100 units $mL^{-1}$ penicillin, and 100 μg $mL^{-1}$ streptomycin (Wako, 168-23191, Tokyo, Japan) using a 100-mm tissue culture dish (Sumitomo, MS-13900, Tokyo, Japan) until the cells reached near-confluency. The cells were detached from the dish by incubating with 2 mL of 0.25% (W/V) trypsin-ethylenediaminetetraacetic acid (EDTA) (Wako, 205-16945, Tokyo, Japan) for 10 min at 37 °C, collected by centrifugation (Hitachi, CF7D2, Tokyo, Japan) for 3 min, and resuspended in a RPMI-1640 medium containing 10% FBS at $1 \times 10^5$ cells $mL^{-1}$. The cells were incubated with 1-mM 5-aminole-vulinic acid (5-ALA) in PBS at 37 °C for 3 h. After incubation, the 5-ALA solution was removed, and the cells were washed with PBS. The collected cells were stained by 1 mL of PBS containing 10 μL of anti-EpCAM antibody (VU-1D9, GeneTex, Irvine, CA). After 30 min of incubation at ambient temperature, the cells were washed with 1 mL of PBS twice and reacted with 2.5 μL of Alexa Flour 488 goat anti-mouse IgG1 (A21121, Invitrogen, Carlsbad, CA) and 2000× diluted Hoechst 33342, H3570 (Invitrogen, Carlsbad, CA) in PBS at ambient temperature for 20 min, followed by washing with 1 mL of PBS twice. The resultant stained cells were resuspended in 1 mL of PBS and stored at 4 °C with light blocked until use.

**Preparation of murine neutrophils and lymphocytes.** C57BL/6 mice were used in this study and purchased from CLEA Japan. All mice were kept under specific pathogen-free conditions. All animal experiments were done in accordance with protocols approved by the Animal Care and Use Committee at The University of Tokyo. The femur bones from C57BL/6 female mice (8 weeks old) were collected by cutting above and below the joints. Bone-marrow cells were washed out of each bone by inserting a needle (26 gauge) with a sterile syringe filled with PBS/2% FBS into one side of the bone. After removing red blood cells by lysis buffer (Sigma-Aldrich), white blood cells were stained with biotinylated anti-Ly6G antibody (RB6-8C5, BioLegend). The cells were secondary-stained with V500-conjugated Streptavidin (BD Biosciences) and Pacific Blue-conjugated anti-CD3ε (145-2C11),

-CD4 (RM4-5), -CD8α (53-6.7), -B220 (RA3-6B2), -NK1.1 (PK136) antibody (BioLegend) were used for FACS analysis. Each antibody was used for staining at the final concentration of 1 μg $mL^{-1}$ in PBS/2% FBS for 30 min on ice, then washed and resuspended in PBS/2% FBS. Neutrophils and lymphocytes were further sorted by FACSAria IIμ (BD Biosciences) as V500 and Pacific Blue single positive cells, respectively, which have no overlap in excitation and emission spectra in the analysis of our imaging flow cytometer. Neutrophils and lymphocytes were suspended in a FBS-free RPMI-1640 culture medium and stained with 1 μM SYTO16 and 10 μM CellTracker Red. The cells were incubated at 37 °C for 45 min, then washed and resuspended in PBS.

**Preparation of E. gracilis cells.** E. gracilis NIES-48 was obtained from the Microbial Culture Collection at the National Institute for Environmental Studies[42]. It was cultured in culture flasks (working volume: 20 mL) using a modified AF-6 culture medium without a dissolved carbon source. The culture was maintained at 25 °C and illuminated in a 14:10-hour light:dark pattern (approximately 120 μmol photon $m^{-2} s^{-1}$). A group of E. gracilis cells was pre-cultured in a modified AF-6 medium. The cells were cultured in the nitrogen-deficient medium for 5 days and denoted as cells in "nitrogen-deficient condition". For the observation of intra-cellular lipid bodies, a stock solution of 1-mM BODIPY505/515 (Thermo Fisher Scientific, USA) in dimethyl sulfoxide containing 1% ethanol was prepared. Both the nitrogen-sufficient and nitrogen-deficient E. gracilis cells (~$10^6$ cells $mL^{-1}$) were stained with 10 μM of BODIPY505/515 in de-ionized water, incubated without light for 30 min, washed, suspended in de-ionized water, and immediately used for imaging.

**Imaging sensitivity.** The imaging sensitivity of VIFFI flow cytometry is evaluated by SNR in comparison with conventional methods. Here, we evaluate the SNR of a fluorescence image obtained by an imaging flow cytometer by using the SNR of the camera readout per pixel. The signal level in the unit of the number of electrons is expressed by

$$S \sim PC^{-1} Tns\eta_{yield}\eta_{img}\eta_{sensor}, \tag{3}$$

where $P$, $C$, $T$, $n$, $s$, $\eta_{yield}$, $\eta_{img}$, and $\eta_{sensor}$ denote the power of the excitation beam, the cross section of the excitation beam, the time of the excitation beam illumination for a single fluorescent molecule, the number of fluorescent molecules in a single-pixel area, the absorption cross section of the fluorescent molecule, the quantum yield of the fluorescent molecule, the photon collection efficiency of the imaging system, and the quantum efficiency of the image sensor, respectively. Assuming that the detection noise consists of shot noise and readout noise, the SNR is given by

$$\frac{S}{N} = \frac{S}{\sqrt{S + \sigma^2}}, \tag{4}$$

where $\sigma$ denotes the readout noise of the image sensor. The SNRs of VIFFI flow cytometry, TDI-based IFC (Luminex ImageStream®X Mark II), and stroboscopic illumination IFC[18] are shown in Supplementary Fig. 8. The parameters used for the calculations are summarized in Supplementary Table 1. Since the values of $\eta_{sensor}$ and $\sigma$ were unknown for ImageStream®X Mark II, we estimated them from the specifications of a commercial TDI-CCD camera (Hamamatsu C10000-801), which has similar specifications of image acquisition with ImageStream®X Mark II. For a fair comparison, we assumed the identical cross section of the excitation beam and 10% lower efficiency in the excitation beam power and the image formation for VIFFI flow cytometry, considering the power loss at the excitation beam scanner, the relay lens systems, and the polygon scanner. As shown in Supplementary Fig. 8, VIFFI flow cytometry has a comparable SNR with the TDI-based IFC at various numbers of fluorescent molecules per pixel, but TDI-based IFC fails to go beyond the flow speed of ~0.04 m $s^{-1}$ (corresponding to a throughput of ~400 cells $s^{-1}$ at an average cell spacing of 100 μm and a pixel size of 0.325 μm) due to the limited readout rate of the CCD (up to ~100 MS $s^{-1}$). Also, in comparison with strobo-scopic illumination IFC, VIFFI flow cytometry provides a significantly higher SNR by more than a factor of 30 due to VIFFI's much longer exposure time, enabling image acquisition with a reasonable SNR even at a high flow speed of ~1 m $s^{-1}$ (corresponding to a throughput of ~10,000 cells $s^{-1}$ at an average cell spacing of 100 μm and a pixel size of 0.325 μm). Consequently, VIFFI flow cytometry is the only method that enables high-SNR fluorescence IFC of cells flowing at a flow speed as high as 1 m $s^{-1}$.

It is important to note that the required SNR strongly depends on the application and the techniques of image processing. Overall, the analysis that uses finer structures of images tends to require higher SNRs. In fact, larger objects or structures in an image are preserved after low-pass filtering, enabling the analysis at an improved SNR. On the contrary, if a target object has a single-pixel size, low-pass filtering decreases the signal level as well as the noise level, resulting in the loss of the fine structure without significantly improving the SNR.

To compare VIFFI flow cytometry with conventional methods in imaging sensitivity, we measured its detection limit in terms of molecules of equivalent soluble fluorochrome (MESF), which is the most commonly used parameter for evaluating the detection sensitivity of imaging and non-imaging flow cytometers. Our results [MESF = ~50 for the green and red channels (ch1 and ch2), Supplementary Fig. 9] indicate the high sensitivity of the VIFFI flow cytometer

even at a high flow speed of 1 m s$^{-1}$. On the other hand, it is important to note that identical measurement conditions (e.g., flow speed, excitation laser power) are required for a fair apple-to-apple comparison between this value and those of other imaging or non-imaging flow cytometers. Unfortunately, while the MESF values of commercial imaging or non-imaging flow cytometers are available, these conditions are not often disclosed in their brochures or specification sheets. Therefore, it is ideal to discuss the SNR shown above along with the MESF for a fair apple-to-apple comparison between different imaging or non-imaging flow cytometers, including the VIFFI flow cytometer.

**Data acquisition speed**. The data acquisition speed of IFC is evaluated by the (effective) line rate. The cell throughput is a common index of the data acquisition speed of flow cytometry, but it may cause confusion in the case of IFC because it is influenced by other parameters such as pixel size and cell spacing. In other words, the cell throughput has trade-off relations with other parameters, complicating the comparison between different IFC systems. Parameters related to the data acquisition speed for VIFFI flow cytometry and TDI-based IFC (ImageStream®X Mark II) are summarized in Supplementary Table 2. The effective line rate is calculated as the flow speed divided by the pixel size in the flow direction and is given by 3.1 MHz and 0.12 MHz for VIFFI flow cytometry and ImageStream®X Mark II (assuming 60-X magnification), respectively. On the other hand, the throughput is proportional to the line rate such that their relation is formulated by $f_{th} = p_x f_x / l_x$, where $f_{th}$, $p_x$, $l_x$, and $f_x$ denote the cell throughput, the pixel size (pixel resolution) in the object plane in the flow direction, the average cell spacing, and the line rate, respectively. The pixel size affects the data volume and information content of a single-cell image. The average cell spacing is determined by sample preparation, not by the instruments. Therefore, for a fair apple-to-apple comparison of different imaging flow cytometers, the cell throughput should be compared under the same conditions of pixel size and average cell spacing. In this manner, the VIFFI flow cytometer has a factor of 26 higher cell throughput than ImageStream®X Mark II.

**Trade-offs between SNR, pixel size, FOV, and flow speed**. The trade-off relations between SNR, the pixel size, FOV, and flow speed are formulated by

$$\frac{FOV_x - l_0}{v} \approx \frac{FOV_x FOV_y}{f_P s_P} + t_{exp1}, \tag{5}$$

$$t_{exp2} = \frac{L_{scan}}{v}, \tag{6}$$

$$\frac{S}{N} \approx \beta \sqrt{\min(t_{exp1}, t_{exp2})}, \tag{7}$$

where $FOV_x$, $FOV_y$, $l_0$, $v$, $f_P$, $s_P$, $t_{exp1}$, $t_{exp2}$, $L_{scan}$, $S$, and $\beta$ denote the FOV in the flow direction, the FOV in the direction perpendicular to the flow direction, the overlapped length between consecutive frames in the flow direction, the flow velocity, the pixel data rate of the sCMOS camera (572 MHz), the pixel area, the upper limit on the exposure time determined by the camera's data transfer speed, the upper limit on the exposure time determined by the maximum scan range in the polygon scanner (the left hand side of Eq. (1)), the maximum scan range on the object, the signal level, and the proportionality coefficient, respectively. Equation (5) represents the frame period. The first term of the right-hand side in Eq. (5) represents the readout time of the camera. Equation (6) gives the constraint on $t_{exp2}$ set by $L_{scan}$, which is determined by the design of the optical imaging system as discussed in Section 1 (see also Supplementary Fig. 2). If we assume an arbitrary design in the optical imaging system, Eq. (6) is negligible. In Eq. (7), we assumed that only the shot noise predominantly contributes to the SNR on the basis of the results in Section 15. For more rigorous calculations accounting for the readout noise, which is important in the case of low SNRs, we may need to modify Eq. (7) on the basis of Eq. (4). Equations (5–7) provide a comprehensive overview of the trade-off relations, as well as their quantification. Supplementary Fig. 3 shows an example of a specific trade-off relation between $FOV_y$ and $v$, which can be derived from Eq. (5).

From a practical point of view, it is convenient to rewrite Eq. (5) using adjustable parameters of a system such as the magnification of the imaging system ($M$) and the number of lines in the direction perpendicular to the flow direction ($N_y$). Some of the above parameters are expressed by

$$FOV_x = \frac{L_x}{M}, \tag{8}$$

$$FOV_y = \frac{N_y d_P}{M}, \tag{9}$$

$$s_P = \frac{d_P^2}{M^2}, \tag{10}$$

where $d_P$ denotes the pixel size of the camera (6.5 μm). Then, Eq. (5) is rewritten as

$$\frac{L_x/M - l_0}{v} \approx \frac{L_x}{f_P d_P} N_y + t_{exp1}. \tag{11}$$

Practically, we adjust the values of $M$ (via the choice of the objective lens), $N_y$ (via the configuration of the camera), and $v$ (via the setting of the flow system) to find the best balance between the sensitivity, FOV in the direction perpendicular to the flow direction, the pixel size, and the throughput using Eq. (6), Eq. (7), and Eq. (11). The spatial resolution is another important parameter of the system that is not considered in the above discussion, but is also adjusted via the choice of the objective lens.

**Spatial resolution**. We evaluated the spatial resolution of the VIFFI flow cytometer using 18,000 images of 200-nm fluorescent beads (Fluoresbrite® YO Carboxylate Microspheres 0.20 μm, Polysciences, Inc.) obtained with excitation laser light at 488 nm. The ratio of the volumetric flow rates of the sample flow and sheath flow was set to 1:3900. We estimated the FWe$^{-1}$M width of the point-spread function (PSF) by fitting Gaussian functions with offset values to the x- and y-profiles of an image of each bead. The results are shown in Supplementary Fig. 15, which characterizes statistical features of the spatial resolution of the VIFFI flow cytometer. First, the difference in PSF distribution between the x and y directions represents the elongation of the PSF due to the residual motion blur of the fluorescent beads, which is less than one pixel (325 nm) on average. Second, the distributions in the y direction with tailed shapes on the right-hand side represent image defocus. Third, the difference in PSF distribution between the red and green channels reflects the wavelength dependence of the wavefront aberration (including defocus) of the VIFFI flow cytometer and that of the diffraction-limited size of the PSF.

**Reporting summary**. Further information on research design is available in the Nature Research Reporting Summary linked to this article.

## Data availability

The source data underlying Fig. 4b, c, Fig. 5b, c, Supplementary Figs. 3, 9, 11–15 is available in our Source Data file. An additional dataset that supports findings in this study is available upon reasonable request to the corresponding authors.

## Code availability

Our image analysis codes are available at https://github.com/MortisHuang/VIFFI-image-analysis (codes for images of cells) and https://github.com/hideharu-mikami/VIFFI-flbeads (codes for images of fluorescent beads).

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

## Acknowledgements

This work was supported by ImPACT Program (CSTI, Cabinet Office, Government of Japan), JSPS Core-to-Core Program, JSPS KAKENHI Grant Number 19H05633, White Rock Foundation, and Precise Measurement Technology Promotion Foundation. We thank ImPACT collaborators for their help with the evaluation of imaging sensitivity.

## Author contributions

H.Mikami and Y.Ozeki conceived the idea. H.Mikami, M.K., and Y.Ozeki designed the setup. H.Mikami, K.M., H.Matsumura built the experimental setup and performed the experiments. H.Mikami, C.-J.H., K.H., T.S., and C.L. performed the data analysis. S.Ueno, T.Miura, T.I., K.N., T.Maeno., H.W., M.Y., S.Uemura, S.O., Y.Ohya, H.K., and S.M. prepared the samples. C.-W.S. supervised C.-J.H.'s computational work. Y.Ozeki and K.G. supervised the project. H.Mikami and K.G. wrote the paper with assistance from all the co-authors.

## Competing interests

H.Mikami, Y.Ozeki, and K.G. are inventors on a pending patent that covers a part of key ideas of VIFFI flow cytometry (PCT/JP2017/031937, applied by The University of Tokyo). T.S. and K.G. are shareholders of CYBO, Inc. The authors declare no other competing interests.
