## [Peer Review File · Nature Communications]

Reviewers' Comments:

Reviewer #1:

Remarks to the Author:

The paper describes a method of using a rotating polygon reflector of controlled speed that corresponds to the travel speed of cells in the microfluidic channel to remove motion blur for imaging flow cytometers. With motion blur correction, the cells can flow at high speed (meters per second) to raise the system throughput. The authors claim that the design can overcome the tradeoffs among sensitivity, throughput, and image resolution. Although the paper contains quite a lot of materials and rather detailed descriptions of the system, the reviewer has major concerns about the real benefits of the system. The major concerns are:

(a) The use of light sheet contributes to the "perceived" spatial resolution enhancement by avoiding the problem of focal depth. However, the attained cell image is simply a slice of randomly chosen cell cross section (of the beam waist of the light sheet). As a result, the attained image data can be misleading. For example, if the light sheet happens to section the equator of the nucleus for one cell and the apex of the nucleus for another cell, the images of the nucleus of the two cells can look very different even though they are actually similar. That is why people in microscopy field always use light sheet to produce 3D images instead of 2D images because the information can be highly biased for the latter.

(b) Although the authors claim enhanced spatial resolution, the paper lacks any quantitative data to show the real spatial resolution. The real spatial resolution can be obtained by showing the image of the point spread function (PSF) by imaging a very small particle. One can then create the histogram of the PSF to "quantify" the effects of imaging optics, pixilation effect, chromatic dispersion (by doing PSF at two wavelengths), and motion blur, etc..

(c) Similarly, to support the claim on enhanced sensitivity, the authors should run the "rainbow bead experiment". Rainbow beads are beads of the same size but different fluorescent intensity and color. A commercial high-sensitivity flow cytometer can resolve 8 intensity peaks for each color in the histogram for rainbow beads. To demonstrate high sensitivity (or sensitivity comparable to the commercial systems), the authors can run through the rainbow beads to show the fluorescent images of beads of different intensity (and color) in a histogram.

There is one relatively minor issue the authors may also want to address:

The system drawing shows one scanning laser light sheet. However, most experiments in the paper use two lasers (488nm and 560nm lasers). The authors may want to revise the schematics to show how the two laser system is designed (i.e. do the two lasers share the same AOD and cylindrical lens or not? How are the two-color images formed (require any signal processing or frame shifting)? The information given in the Methods is a bit hard to follow without the help of schematics.

Reviewer #3:

Remarks to the Author:

Review of Virtual-freezing fluorescence imaging flow cytometry by Mikami et al

In the manuscript entitled "virtual-freezing fluorescence imaging flow cytometry" by Mikami et al, the authors describe a new approach for addressing the issues between speed of image acquisition and image quality in the field of Imaging Flow Cytometry (IFC). As the authors correctly state, there is a negative correlation presently between throughput of image acquisition and the quality of said images. IFC is indeed a very powerful tool and allows for the measurement and classification of cell biology at a level not possible however the scalability of the technology is grossly hampered by the number of cell events it can capture compared to zero-resolution classical

flow cytometry. Moreover, speeding the technology up simply to collect poor quality images greatly affects the ability to measure and resolve the biology on a per cell basis. Several groups have attempted to address this issue but seem more focused on throughput rather than image quality. The reality of any image-based flow approach is that while throughput is important for scalability issues, the image quality is more important as it will "make or break" the ability to derive resolving features for different cell types. The technology developed by Mikami et al does seem to have addressed this issue of throughput at the expense of image quality by introducing a clever "virtual freezing" approach. Overall I feel that this is an excellent body of work, and may finally answer the longstanding issues of throughput versus image quality that has frankly hampered the field of IFC. I have read the manuscript and supplemental material several times and I genuinely struggle to find any major faults with the work. I am particularly impressed by the sheer body of highly relevant biological data that they authors present in order to showcase the application of their technology. The image quality and thus feature-based resolution of different biological states seems convincing. The authors also present excellent fluorescence sensitivity as shown by 8 peak beads (suppl. Fig 9) of around 50 MESF for both channels, bringing it close to conventional non-imaging flow cytometry. My only minor comment would be that the discussion is lacking and could cover more in terms of future applications and where next for VIFFI. To me this is a significant advancement in the field of fast cell imaging in flow so the discussion should really do better to position this technology better. Otherwise I would like to commend the authors for an excellent study that will greatly advance the field.

We are grateful to the Reviewers for taking the time to review our manuscript and providing us with their valuable comments. We have taken all of the comments into consideration and have made appropriate changes to the manuscript. Our point-by-point response appears below, in which we first echo each Reviewer's comments (shown in italic) and then respond to them. Our revisions are shown in the revised manuscript **in red**.

To Reviewer 1:

Reviewer 1's comment #1:

The paper describes a method of using a rotating polygon reflector of controlled speed that corresponds to the travel speed of cells in the microfluidic channel to remove motion blur for imaging flow cytometers. With motion blur correction, the cells can flow at high speed (meters per second) to raise the system throughput. The authors claim that the design can overcome the tradeoffs among sensitivity, throughput, and image resolution. Although the paper contains quite a lot of materials and rather detailed descriptions of the system, the reviewer has major concerns about the real benefits of the system.

Authors' response:

We thank the Reviewer for understanding the significance of our manuscript.

Reviewer 1's comment #2:

(a) The use of light sheet contributes to the "perceived" spatial resolution enhancement by avoiding the problem of focal depth. However, the attained cell image is simply a slice of randomly chosen cell cross section (of the beam waist of the light sheet). As a result, the attained image data can be misleading. For example, if the light sheet happens to section the equator of the nucleus for one cell and the apex of the nucleus for another cell, the images of the nucleus of the two cells can look very different even though they are actually similar. That is why people in microscopy field always use light sheet to produce 3D images instead of 2D images because the information can be highly biased for the latter.

Authors' response:

We thank the Reviewer for the comment. The Reviewer is correct about the negative aspect of light-sheet excitation. In fact, we are very much aware of it and have designed the VIFFI flow cytometer to mitigate it. We feel that it is important to note that the purpose of the light-sheet illumination in this work is to maximize the imaging sensitivity of the VIFFI flow cytometer, not to obtain sectioned images of cells. In fact, the thicknesses of the excitation beams (as described in "Constructed optical setup" in the Methods section) are designed to be comparable to the focal depth of the VIFFI system so that the light-sheet illumination does not affect the depth resolution significantly. Furthermore, while 2D imaging may fail to accurately capture the morphological features of cells of some particular species (which is a common issue in all types of 2D imaging flow cytometry and is not unique to VIFFI flow cytometry), this issue can be overcome by employing an extended depth-of-field (EDF) technique. In fact, a commercial imaging flow cytometer (Luminex ImageStream^{®X} Mark II) has an EDF option based on a cubic phase modulation technique. While 3D imaging flow cytometry may be an attractive option, unfortunately, the cell throughput needs to be sacrificed significantly due to increased data size per cell. Therefore, we believe that our method is effective for applications that simultaneously require high throughput, high sensitivity, and high spatial resolution while 3D imaging flow cytometry is useful for applications that require volumetric imaging of cells at the expense of throughput. High-throughput 3D imaging flow cytometry would be an ideal goal for many researchers, but is outside the scope of this manuscript. To address the Reviewer's comment, we have added text about the issue and how to solve it to the Discussion section of the revised manuscript.

Reviewer 1's comment #3:

(b) Although the authors claim enhanced spatial resolution, the paper lacks any quantitative data to show the real

spatial resolution. The real spatial resolution can be obtained by showing the image of the point spread function (PSF) by imaging a very small particle. One can then create the histogram of the PSF to “quantify” the effects of imaging optics, pixilation effect, chromatic dispersion (by doing PSF at two wavelengths), and motion blur, etc..

Authors’ response:

We thank the Reviewer for the comment and suggestion. We think the original manuscript had quantitative data for the spatial resolution (Supplementary Fig. 13 and Supplementary Fig. 14) which we think would partly address the comment. However, in order to further clarify the quantified spatial resolution, we have added a new paragraph and a new supplementary figure (now Supplementary Fig. 15) to the Methods section and the Supplementary Information, respectively. To respond to the Reviewer’s suggestion, we have performed statistical analysis of FWe^{-1}M PSF widths using 18,000 images of 200-nm fluorescent beads (the smallest beads with reasonable fluorescence strength in both the green and red detection channels). The results are also shown in Supplementary Fig. 15 in the revised Supplementary Information file.

Reviewer 1’s comment #4:

(c) Similarly, to support the claim on enhanced sensitivity, the authors should run the “rainbow bead experiment”. Rainbow beads are beads of the same size but different fluorescent intensity and color. A commercial high-sensitivity flow cytometer can resolve 8 intensity peaks for each color in the histogram for rainbow beads. To demonstrate high sensitivity (or sensitivity comparable to the commercial systems), the authors can run through the rainbow beads to show the fluorescent images of beads of different intensity (and color) in a histogram.

Authors’ response:

We thank the Reviewer for the comment. We believe that the data shown in Supplementary Fig. 9 in the original manuscript directly addresses the comment. We have demonstrated the high sensitivity with the detection limits of ~50 MESF for both the green and red channels, which are comparable to the sensitivity of commercial (non-imaging) flow cytometers.

Reviewer 1’s comment #5:

The system drawing shows one scanning laser light sheet. However, most experiments in the paper use two lasers (488nm and 560nm lasers). The authors may want to revise the schematics to show how the two laser system is designed (i.e. do the two lasers share the same AOD and cylindrical lens or not? How are the two-color images formed (require any signal processing or frame shifting)? The information given in the Methods is a bit hard to follow without the help of schematics.

Authors’ response:

We thank the Reviewer for the suggestion. We believe that Supplementary Fig. 1 and Supplementary Fig. 10 directly address the comment. As Supplementary Fig. 1 shows, the two lasers share the same AOD and cylindrical lens. The two-color channels are split in the flow direction by a relative tilt angle between the dichroic mirror and the reflection mirror in front of the tube lens.

To Reviewer 3:

Reviewer 3’s comment #1:

In the manuscript entitled “virtual-freezing fluorescence imaging flow cytometry” by Mikami et al, the authors describe a new approach for addressing the issues between speed of image acquisition and image quality in the field of Imaging Flow Cytometry (IFC). As the authors correctly state, there is a negative correlation presently between throughput of image acquisition and the quality of said images. IFC is indeed a very powerful tool and allows for the measurement and classification of cell biology at a level not possible however the scalability of the

technology is grossly hampered by the number of cell events it can capture compared to zero-resolution classical flow cytometry. Moreover, speeding the technology up simply to collect poor quality images greatly affects the ability to measure and resolve the biology on a per cell basis. Several groups have attempted to address this issue but seem more focused on throughput rather than image quality. The reality of any image-based flow approach is that while throughput is important for scalability issues, the image quality is more important as it will “make or break” the ability to derive resolving features for different cell types. The technology developed by Mikami et al does seem to have addressed this issue of throughput at the expense of image quality by introducing a clever “virtual freezing” approach. Overall I feel that this is an excellent body of work, and may finally answer the longstanding issues of throughput versus image quality that has frankly hampered the field of IFC. I have read the manuscript and supplemental material several times and I genuinely struggle to find any major faults with the work. I am particularly impressed by the sheer body of highly relevant biological data that they authors present in order to showcase the application of their technology. The image quality and thus feature-based resolution of different biological states seems convincing. The authors also present excellent fluorescence sensitivity as shown by 8 peak beads (suppl. Fig 9) of around 50 MESF for both channels, bringing it close to conventional non-imaging flow cytometry.

Authors’ response:

We thank the Reviewer for giving us the very positive comment and understanding the significance of our work.

Reviewer 3’s comment #2:

My only minor comment would be that the discussion is lacking and could cover more in terms of future applications and where next for VIFFI. To me this is a significant advancement in the field of fast cell imaging in flow so the discussion should really do better to position this technology better.

Authors’ response:

We thank the Reviewer for the suggestion. We have revised the discussion section in the manuscript to discuss future applications of our method as suggested by the Reviewer.

Reviewers' Comments:

Reviewer #1:

Remarks to the Author:

The responses are satisfactory and the paper can be published as is.